

# Catchment export of base cations: Improved mineral dissolution kinetics influence the role of water transit time

Martin Erlandsson Lampa[1,2,*], Harald U Sverdrup[3], Kevin H Bishop[4], Salim Belyazid[5], Ali Ameli[6],
Stephan J Köhler[4]

[1]Department of Earth Sciences, Uppsala University, Villavägen 16, 752 36 Uppsala, Sweden
[2]Department of Physical Geography and Ecosystem Sciences, Lund University, Sölvegatan 12, 223 62 Lund, Sweden
Sweden
[3]Faculty of Industrial Engineering, Mechanical Engineering and Computer Science, University of Iceland, Sæmundargata 2,
101 Reykjavík, Iceland
[4]Department of Aquatic Sciences and Assessment, Swedish University of Agricultural Sciences, Box 7050, 750 07 Uppsala,
Sweden
[5]Department of Physical Geography, Stockholm University, 106 91 Stockholm, Sweden
[6]Department of Earth, Ocean and Athmospheric Sciences, University of British Columbia, 2020 - 2207 Main Mall, Vancouver,
British Columbia V6T 1Z4, Canada

*Current address: Vattenmyndighetens kansli, Länsstyrelsen i Västmanlands län, 721 86 Västerås, Sweden

**Correspondence**: Martin Erlandsson Lampa (martin.erlandsson.lampa@lansstyrelsen.se)

**Abstract.** Soil mineral weathering is one of the major sources of base cations (BC), which play a dual role for a forest ecosystem; they function both as plant nutrients, and for buffering against acidification of catchment runoff. On a long-term basis, the soil weathering rates will determine the highest sustainable forest productivity without causing acidification. It is believed that the hydrologic residence time plays a key role in determining weathering rates on a landscape scale.

The weathering model PROFILE has been used for almost 30 years to calculate weathering rates in the rooting zone of forest soils. However, the mineral dissolution equations in PROFILE are not adapted for the unsaturated zone, and employing these equations on a catchment scale results in a significant over-prediction of base cation release rates to surface waters. In this study we use a revised set of PROFILE equations which, among other features, include retardation from silica concentrations. Relationships between the water transit time (WTT) and soil water concentrations were derived for each base cation, by
simulating the soil water chemistry along a one-dimensional flowpath, using the mineralogy from a glacial till soil. We show how the revised PROFILE equations are able to reproduce patterns in BC- and Si-concentrations, as well as BC-ratios ($Ca^{2+}/BC$, $Mg^{2+}/BC$ and $Na^+/BC$), observed in soil water profiles and catchment runoff. As opposed to the original set of PROFILE equations, the revised set of equations could reproduce how increasing WTT led to decreasing $Na^+/BC$, as well as increasing $Ca^{2+}/BC$ and $Mg^{2+}/BC$. Furthermore, the total release of base cations from a hillslope was calculated using a mixing
model, where water of different WTT was mixed according to an externally modelled WTT-distribution. The revised set of





equations gave a 50% lower base cation release (0.23 eq·m-2·yr-1) than the original PROFILE equations, and are in better agreement with mass balance calculations of weathering rates. The results from this study thus demonstrate that the revised mineral dissolution equations for PROFILE are a major step forward in modelling weathering rates on a catchment scale.

## 1 Introduction

The dissolution of minerals in soils is one of the key processes creating the diversity of surface water chemistry around the world. In forest ecosystems, mineral dissolution is a major source of base cations. These are tree nutrients in soil water and provide buffering capacity to both soil and the runoff sustaining aquatic ecosystems. Both forest harvest and acid deposition are anthropogenic factors the remove base cations, with the potential to create imbalances that threaten soil and aquatic ecosystems (Akselsson et al., 2007). Quantifying the base cation release rates from mineral dissolution is therefore fundamental

when determining sustainable rates of forest harvesting, as well as critical loads of acidifying deposition (Klaminder et al., 2011). Extensive modeling efforts have been made to define sustainability with respect to tree health (Blum et al., 2002; Gerard et al., 2008; Akselsson et al., 2016), however, there are only a few examples of explicitly modelling mineral dissolution on a catchment scale (i.e. Goddéris et al., 2006) in an effort to determine the long term sustainability of ecosystems. Instead key management decisions for controlling acid deposition and forest management practices have been based on approaches that

do not explicitly predict weathering rates.

One key influence on any process-based understanding of base cation flows in soils is a clear relationship between water transit time (WTT) and the amount of weathering products exported from the catchment – the longer time the soil water is in contact with mineral surfaces, the more solutes will be released to pore water that can eventually exit the catchment as runoff (Maher,

2010). If chemical weathering dominates over other base cation flux terms (biological uptake, deposition etc.), the relationship between WTT and solute concentrations in runoff will be linear as long as minerals dissolve at a steady rate. This means that increasing the water flux dilutes the concentration seen in soil waters and stream runoff, creating a negative relationship between stream concentration and discharge. However, as the concentrations of weathering products build up in the soil water solution, the mineral dissolution rates will slow down. This is a part of the complexity of predicting concentration-discharge

relationships that can often include chemostatic behavior (Godsey et al., 2009; Maher 2011). As minerals react individually to changes in the surrounding solution, this shift may be timed differently for different solutes for different flux rates, all operating in a catchment system where the mineralogy and physical properties will change over time in response to dissolution processes, secondary mineral formation and other processes.

Process based models of soil weathering are powerful tools with the potential to couple hydrological processes and mineral dissolution rates. The steady-state weathering model PROFILE (Warfvinge and Sverdrup, 1992), and its dynamic successors SAFE (Sverdrup and Warfvinge, 1993) and ForSAFE (Wallman et al., 2006), have consistently proven capable of reproducing



observations of field conditions in the rooting zone over a wide range of soil types and climate zones (Sverdrup et al., 1995; Kolka et al., 1996; Langan et al., 1995; Sverdrup et al., 1998; Zabowski et al., 2007; Phelan et al., 2014). However, PROFILE was developed for and is restricted to the unsaturated soil domain, and can furthermore only handle vertical flow. Application of the PROFILE equations for mineral dissolution rates to the saturated zone (Erlandsson et al., 2016), as well as initial efforts

to use ForSAFE for two dimensional simulations of hillslope weathering (Zanchi, 2016), revealed significant over-prediction of weathering rates below the water table, and over-estimates of $Na^+$ and $K^+$-release to the stream, suggesting shortcomings in some aspects of the model.

Aside from differences in the mineral dissolution process, prediction of catchment weathering rates also demands a better

understanding of hydrological processes. Catchment hydrology adds process complexity beyond that encountered in the unsaturated zone, requiring descriptions of both vertical and lateral flow. There are also still unresolved issues regarding transit time distributions, flow paths and "immobile water" (McDonell and Beven, 2014; Pinay et al., 2015). For the weakly buffered glacial till soils covering large parts of the boreal/nemoral vegetation zones, the soil and surface water systems have base cation dissolution rates where acid deposition and/or forest harvest can significantly affect the buffering capacity and nutrient supply

(Akselsson et al., 2019). Due to the strong gradient in soil physical properties, these systems have been problematic to model hydrologically (Amvrosiadi et al., 2017a). Recently, advances have been made in the development of models able to simulate both runoff and transport of conservative elements from catchments (Ameli et al., 2017). While the theoretical importance of correctly defining both the water flow system and the chemical dissolution rates has been identified (Maher, 2010), it remains to be seen how the dissolution rates of specific minerals found in shallow till systems of weakly buffered glacial till soils

interact with possible distributions of water fluxes in these environments.

It has previously been known that the current PROFILE mineral dissolution equations do not form a complete description of the dissolution process (Rapp and Bishop, 2003). Specifically, the retarding effect from aqueous silica, a major constituent of silicate minerals, has previously not been included. While this is less important in the unsaturated zone where aqueous silica

concentrations are low (typically < 0.3mM), it has been suggested that the over-prediction of weathering in the saturated zone is caused by neglecting the silica effect. In this issue (Sverdrup et al., 2019), a revised set of PROFILE equations is presented which contain a number of expanded functions, including retardation from silica concentrations. In the present study, we use both the original and the revised PROFILE equations to evaluate how the mineral dissolution rates interact with the water transit time. We utilize a simple 1D reactive-transport model, simulating the dissolution of minerals and the evolution of soil

water chemistry along a one-dimensional flowpath created to simulate the movement of a water particle through the saturated zone of a catchment. Given mineralogy and mineral surface area of the soil, the model will produce a unique relationship between WTT and solute concentration for each individual base cation. We then evaluate the results against soil- and stream water data, specifically against how the ratios between different base cations ($Ca^{2+}$, $Mg^{2+}$, $Na^+$, $K^+$) vary with different proxies for WTT. Furthermore, we demonstrate how the base cation release rates from mineral dissolution in a hillslope comprised of



a range of flow paths with different WTT can be calculated by mixing the water from the mineral dissolution model according to a given WTT-distribution.

**2 Material and methods**

**2.1 Dissolution rate equations**

*2.1.1 Dissolution rate equations in the PROFILE model*

The mineral specific dissolution rates can generally be written as:

$$r = g(\Delta G_R) \cdot \sum_i F_i(T) \cdot r_{+,i} \qquad \text{(eq. 1)}$$

Where $r_{+,i}$ stands for the forward dissolution rate of the $i$:th reaction, and is a function of the surrounding fluid chemistry, $F_i(T)$

represents a temperature function for the $i$:th reaction and $g(\Delta G_R)$ denotes a chemical affinity function.

Forward dissolution rates in the PROFILE model are calculated as the sum of a number of parallel reactions (Sverdrup 1990, Warfvinge and Sverdrup 1992, Sverdrup and Warfvinge 1993, Warfvinge and Sverdrup 1995):

$$r = \sum k_j(T) \cdot \frac{\{j\}^n}{f_j} \qquad \text{(eq. 2)}$$

Where $j$ is the dissolution agent, $k$ is a temperature dependent reaction coefficient, and $f$ is a so-called "brake"-function, which describes a retarding effect on mineral dissolution. In the original version of PROFILE, the terms for four dissolution agents are defined: $H^+$, $H_2O$, $CO_2$ and one organic anion analogue ($R^-$). A fifth dissolution agent, $OH^-$, is known to be potentially important, but has so far been omitted as it believed not to contribute significantly to mineral dissolution in the unsaturated zone.

The retarding functions are products of factors defined by concentrations of aquatic species with a retarding effect on mineral dissolution:

$$f_j = \prod g_k \qquad \text{(eq. 3)}$$

where $k$ is an aqueous species which retards the dissolution reaction. Defined in the original version of PROFILE are retarding

effects from base cations (BC) and aluminum (Al), and in addition also a self-inhibiting factor for the reaction with $R^-$. There are no brake-functions which apply to the $CO_2$-term, and BC- and Al-brakes are missing from the $R^-$-term.

*2.1.2 Features of the new PROFILE model*

The new version of the PROFILE equations contains several new features (Sverdrup et al., 2019):

1. A new term representing the reaction with $OH^-$ was added to eq. 2.



2. A "Si-brake", to define the retarding effects from aqueous siliceous acid (referred to as "Si" further on), was applied to all terms. This brake takes the following form:

$$g_{Si} = \left(1 + K_{Si} * \left(\frac{[Si]}{C_{Si}}\right)^{z_{Si}}\right) \qquad \text{(eq. 4)}$$

Where $K_{Si}$ is the silica brake constant, $C_{Si}$ is the lower limiting silica concentration and $z_{Si}$ is the silica brake order. This term

will cause weathering rates to decrease with increasing silica concentrations.

3. A $CO_2$-brake was applied to the $CO_2$-term to define a saturation limit of the reaction:

$$g_{CO2} = (1 + K_{CO2} * pCO_2)^{z_{CO2}} \qquad \text{(eq. 5)}$$

Where $K_{CO2}$ is the $CO_2$-brake constant and $z_{CO2}$ is the $CO_2$ brake order (which is equal to $n_{CO2}$, i.e. the reaction order for the

$CO_2$-reaction). This term will avoid an unrealistic rise in weathering at high $CO_2$ soil water concentrations.

4. BC- and Al-brakes were applied to terms for which they were not defined in the original PROFILE equations, i.e. the $CO_2$-term, the R$^-$-term and the OH$^-$-term. The parameters of the new terms can be found in Sverdrup et al. (2019). The new PROFILE equations require aqueous Si to be modelled, in order to make the Si-brakes work properly. Si is known to be retained in the

soil, although it is unclear exactly in which form, and the precipitated phases are often poorly identified (Yang and Steefel, 2008). In this study Si and Al were controlled by letting the soil water solution equilibrate against a "kaolinite-like"-phase.

$$Al_x Si_y O_z (OH)_{(3x+4y-2z)} + 3xH^+ \Leftrightarrow xAl^{3+} + ySiO_2 + (3x + 2y - z)H_2O \qquad \text{(eq. 6)}$$

The stoichiometric coefficients $x$, $y$ and $z$, and the equilibrium constant were then calibrated so that the modelled Si-WTT-relationship resembled the observed Si-depth-relationship in the soil profile S22 at the study site (see section 2.3.3 below).

Besides this reaction, which removes Al and Si from the soil water solution by precipitation, no secondary phases were considered. In comparison with Al and Si, the soil retention of base cations is small.

## 2.2 Model description

The model is a simplified representation of a one-dimensional flowpath through the saturated zone of a hillslope, where the

release of base cations is calculated by simulating how the soil solution chemistry evolves as mineral grains dissolve and precipitate. All calculations were performed with the geochemical code PHREEQC.

The transport through the column was simulated as an advective process. The column was composed of 40 cells, each 1 m in length, thus with a total length of 40 m. Each cell contains (by default in PHREEQC) 1 dm$^3$ of water, and the total cell volume

is thus equal to 1/porosity [dm$^3$]. In each time step, the water in the cell interacts with a specified area of mineral surfaces, which dissolve according to a set of dissolution rate-laws. The model was run for transit times ranging from 1 month to 48 years until steady-state was reached. All column cells are uniform with respect to mineralogy, soil texture, flow velocity,




organic carbon content and carbon dioxide pressure. Ion exchange processes and plant uptake are not included in the model. As the study site is a mature forest with low historic acid deposition, it is likely that the system is close to steady-state with respect to base cation uptake and net exchange.

In the model set-up, precipitation was used as the infilling solution, with a chemical composition calculated as the measured annual bulk deposition (average values years 1999-2011, Zetterberg et al., 2014) divided by the annual runoff. Calculated base cation concentrations in precipitation were: 0.016 meq [$Ca^{2+}$], 0.01 meq [$Mg^{2+}$], 0.019 meq [$Na^+$] and 0.006 meq [$K^+$].

For each cell and time step, the net release of base cations from mineral dissolution ($w$) of each mineral $m$ was calculated
using:

$$w_m = r_m \cdot SA_m \qquad\qquad (\text{eq. 7})$$

Where $r$ is the specific dissolution rate, and $SA$ is the surface area for mineral $m$.

Assuming that the surface areas of different minerals are proportional to their weight fractions, $SA_m$ was calculated according
to the formula:

$$SA_m = \varphi_m \cdot A_{min} \cdot \delta_{min} \cdot V_u \cdot (1 - \emptyset) \cdot \left(1 - \frac{1}{1 + \frac{\delta_{org} - LOI \cdot \delta_{org}}{LOI \cdot \delta_{min}}}\right) \qquad\qquad (\text{eq. 8})$$

where $\varphi_m$ is the weight fraction of mineral $m$, $A_{min}$ is the total field surface area per kg of minerals, $\delta_{min}$ is the mineral density, $\delta_{org}$ is the density of organic material, $V_u$ is the unit volume of one model cell, $\emptyset$ is the porosity and $LOI$ is loss-on-ignition, i.e. the proportion of organic material. The detailed derivation of this equation can be found in Erlandsson et al. (2016).

### 2.3 Site description and field data

#### 2.3.1 Site description

All field data used in the study was taken from the Krycklan Catchment Study (Laudon et al., 2013). The catchment is located in Northern Sweden (64°14'N, 19°46'E), 60 km inland from the Baltic Sea. The area of the whole Krycklan catchment is 67
km², and the elevation ranges from 405 to 114 m.a.s.l. The soil has developed predominantly on glacial till, with larger areas of fluvioglacial deposits at lower altitudes. The vegetation is dominated by coniferous forests (87 %) and mires (9 %).

#### 2.3.2 Data for input to the model

Input data required to run the model includes: mineralogy, estimated surface area per kg minerals, porosity and loss-on-
ignition. These were all taken from observations at 90 cm depth of a soil profile (S22) located 22 m from the headwater stream C2 on the "S-transect" hillslope study that was established in 1996 (Nyberg et al. 2001). Porosity equaled 38 %, and loss-on-



ignition 0.5 %. The average observed mineralogy has the following composition: 48 % quartz, 27 % plagioclase, 19 % K-feldspar, 4.8 % hornblende, 0.94 % pyroxene, 0.70 % biotite and 0.24 % apatite. The mineral surface area was calculated from the soil texture:

$$A_{min} = 0.3 \cdot \chi_{sand} + 2.2 \cdot \chi_{silt} + 8.0 \cdot \chi_{clay} \qquad \text{(eq. 9)}$$

The resulting mineral surface area ($A_{min}$) calculated for 90 cm depth was 966 m$^2 \cdot$kg$^{-1}$, which is much higher than at other depths in the profile. Patterns of base cation concentrations in the depth profile also indicated that there may be an anomaly at 90 cm depth. Therefore, the mineral surface area was instead set to 400 m$^2 \cdot$kg$^{-1}$, similar to observations at 35-75 cm depths. See Erlandsson et al., 2016 for details on calculation of these input data and analytic methods.

### 2.3.3 Data for model evaluation

Three different data sets from the Krycklan Catchment Study were used to compare the modelled relations between WTT and base cation concentrations, as well as base cation ratios (Ca$^{2+}$: Mg$^{2+}$: Na$^+$: K$^+$), with field observations:

15    1. Soil water samples from the S22 soil profile. Samples from the upper meter of soil (12, 20, 35, 50, 75 and 90 cm) are part of a soil water monitoring program where soil water has been sampled regularly for 2004-2012. In addition, we also included samples from deeper groundwater (2.25, 5.2 and 10.7 m) collected from piezometers. Deeper groundwater samples were only collected once, in August 2012. Soil water chemistry was plotted vs. depth as a proxy for WTT.

20    2. Stream water samples from regularly monitored sites of the Krycklan catchment. A total of 13 streams with catchment areas between 0.12 and 67 km$^2$ were included in the analysis (Köhler et al., 2014). The average stream chemistry from samples taken in 2016 was plotted vs. catchment area as a proxy for water transit time. The mineralogy is largely homogeneous across the whole 67 km$^2$ catchment (Ledesma et al., 2013). Large parts (80 %) of the catchment are covered by glacial till similar to the S22 profile, however, the valleys of the lower parts are covered with finer-textured fluvioglacial sediments.

3. Stream water samples from the C2 stream. Samples were collected fortnightly to monthly from March 2003 to December 2012, but more frequently before and after spring flood (> 50 % of the samples were collected during April and May). Samples were taken approximately 400 m downstream from the intersection between the stream and the S-transect. At the sampling point, the catchment area is approximately 0.12 km$^2$, and completely forested on mineral soils, with the exception of a riparian

30    band of histosols extending 2-10 m from the stream. Stream chemistry was plotted vs. decreasing flow as a proxy for increasing WTT.

### 2.3.4 Implementation of dissolution rate-laws

The model was run with three different combinations of dissolution rate laws and mineralogies:





Model 1: Original PROFILE dissolution rate equations.

Model 2: New PROFILE dissolution rate equations

Model 3: Same as Model 2, but assuming that plagioclase consists of one $Ca^{2+}$-free phase (100 % albite), and one $Ca^{2+}$-rich phase (bytownite, composed of 80 % anorthite and 20 % albite), which dissolves independently.

### 2.4 Hydrological modelling of the hillslope

Here, we assumed that the hillslope hydrological flow can be represented as a series of steady-state non-interacting flowpaths with different water transit times and fluxes. We used the water transit time distribution for the study site, modeled in Ameli et al. (2016b) using the integrated flow and transport model developed in Ameli et al. (2016a). The model explicitly calculates
the pathway (and its corresponding transit time and hydrological flux) of each water particle from land surface to stream, for an exponentially depth-decaying pattern of saturated hydraulic conductivity. These water transit times and fluxes were used to calculate the delivery of base cations along different pathways toward the stream.

Indeed, we calculated the base cation transport carried by each "water parcel" of the water transit time distribution. This will
help to demonstrate the relative importance of old groundwater for base cations transport to the stream.

### 3 Results

### 3.1 Field data

#### 3.1.1 Base cation concentrations

Three different data sets of field observations were analyzed, each one reflecting the relationship between base cation concentrations and WTT, as approximated by soil depth, catchment size or stream discharge. Base cation concentrations increased with increasing estimated WTT in all three sets of samples. In the soil profile, soil water base cation concentrations were approximately constant with depth within the unsaturated zone, at around 0.2 meq·$L^{-1}$. Below 75 cm in the permanently saturated zone, base cation concentrations increased with depth, and eventually reached a concentration of 1.2 meq·$L^{-1}$ at a
depth of 10.7 m. Hydrological modeling has estimated the age of this water below the minimum groundwater level as years to a decade or two (Ameli et al., 2016b; Amvrosiadi et al., 2017a)

In the catchment stream water samples, base cation concentrations generally increased with increasing catchment size. Defining a trend by linear regression, the BC concentrations increased from around 0.16 meq·$L^{-1}$ in the headwater stream with
the smallest catchment (0.12 $km^2$) to around 0.30 meq·$L^{-1}$ at the outlet of the Krycklan basin with the largest total catchment area (67 $km^2$).

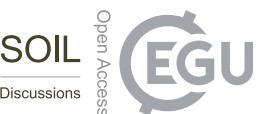

As for the headwater stream C2, the base cation concentrations were lowest at the highest flows. The concentrations increased gently as flow decreased, by about 0.021 meq·L$^{-1}$ per log-unit of runoff (mm/day), down to a runoff of approximately 0.22 mm/day. With a further decrease in runoff, the base cation concentrations increased more sharply, by 0.26 meq·L$^{-1}$ per log-

unit of runoff.

### 3.1.2 Base cation ratios

In the soil profile, all base cation-ratios varied substantially with soil depth. The fraction Mg$^{2+}$/BC increased almost monotonically, from a value of 10 % in the shallowest soil layer to around 25 % in the deepest soil layer. Ca$^{2+}$/BC varied

somewhat irregularly, but increased substantially with depth below the groundwater surface, from 33 % to 54 % in the deepest soil layer. Na$^{+}$/BC increased from around 40 % to 48 % with depth in the unsaturated zone, and then decreased substantially with depth in the saturated zone, to a value of 16 % in the deepest soil layer. K$^{+}$/BC was highest in the shallowest soil layer at 20 %, then dropped sharply to a value of just 3.6 % in the next soil layer, and then displayed an irregular variation with depth. In the stream water samples, in-stream Na$^{+}$/BC decreased significantly with catchment size, from 32 % in the smallest stream

to 22 % in the largest, whereas K$^{+}$/BC increased significantly, from 3.4 % in the smallest stream to 8.2 % in the largest. Ca$^{2+}$/BC and Mg$^{2+}$/BC tended to increase with catchment size, but not significantly.

In the headwater samples, Na$^{+}$/BC and Ca$^{2+}$/BC mirrored each other when plotted against the logarithmic discharge, with Na$^{+}$/BC displaying a maximum value and Ca$^{2+}$/BC displaying a minimum value at a runoff of approximately 0.6-0.7 mm/day.

K$^{+}$/BC displayed a minimum, similar to that of Ca$^{2+}$/BC, but less pronounced, whereas Mg$^{2+}$/BC displayed little systematic variability with runoff.

### 3.2 Model evaluation

#### 3.2.1 Water transit time – solute concentration relationships

Using the original PROFILE dissolution rate equations (model 1), the relationships between WTT and base cation concentrations were nearly linear for all base cations. After four years of WTT, the total base cation concentration at the column outlet equaled 0.76 meq·L$^{-1}$.

Using the new PROFILE dissolution equations (model 2), the relationship between WTT and total base cation concentration

started at the same trajectory as that calculated from the original equations, but it soon begun to deviate from the linear pattern. After one year of WTT, the total BC concentrations calculated from the new set of equations were 80 % of those calculated with the original equations, and after four years, they were 55 % of those from the original equations.



As for the individual base cations, the difference between the two sets of equations was largest for $K^+$, due to the strong Si-brake for K-feldspar. Using the new set of equations, $K^+$ showed an almost chemostatic behavior already after a few months of WTT. $Mg^{2+}$, on the other hand, displayed an almost linear relationship between WTT and solute concentrations for both sets of mineral dissolution equations, due to the weak Si-brakes for all Mg-bearing minerals. $Na^+$, and to a lesser degree $Ca^{2+}$,

were affected by the relatively strong Si-brake for plagioclase. For these two base cations, the WTT-solute concentration relationship derived from the new set of equations begun to deviate from the linear after c:a one year of WTT.

### 3.2.2 Model evaluation - silica and base cation concentrations

For the new PROFILE equations (model 2), the stoichiometry and the equilibrium constant for the secondary "kaolinite-like"

phase was calibrated so the modelled Si-concentrations approximately matched those of the deep soil profile. The equilibrium between the soil solution and the secondary phase could be written as:

$$Al_{1.45}Si_{2.9}O_{3.375}(OH)_{9.2} + 4.35H^+ \leftrightarrow 1.45Al^{3+} + 2.9SiO_2 + 6.775H_2O$$

The calibrated equilibrium constant for this reaction was 104.3.

Using these assumptions, the modelled Si-concentrations were within the same range and showing similar trends with increasing WTT as observed in the deep soil profile. A similar comparison between modelled base cation concentrations plotted against WTT, and base cation concentrations from the soil profile plotted against depth also show a good agreement (Fig 4).

### 3.2.3 Model evaluation - base cation ratios and water transit time

Using the original set of PROFILE equations (model 1), the trends of base cation ratios with increasing WTT were generally opposite to those observed in field data (Fig. 5). $Ca^{2+}$/BC and $Mg^{2+}$/BC both decreased with WTT, from values in the infilling solution of 31 % and 20 % respectively, to end-point values of 25 % and 11 % after 42 years. $Na^+$/BC and $K^+$/BC increased with WTT from 37 % and 11 % in the infilling solution to end-point values of 47 % and 17 % respectively.

With the new set of PROFILE equations (model 2), modelled base cation ratios were more generally in agreement with those observed in field data. $Ca^{2+}$/BC and $Mg^{2+}$/BC both increased, but the latter increased more sharply, and after 42 years WTT, $Mg^{2+}$-concentrations were higher than $Ca^{2+}$-concentrations, which was never seen in field data (Fig. 5). End-point ratios were 38 % for $Ca^{2+}$/BC and 42 % for $Mg^{2+}$/BC. $K^+$/BC decreased gently to an end-point value of 3 %, and $Na^+$/BC decreased sharply

to an end-point value of 17 %.

With the modification of separating plagioclase into a $Na^+$-rich phase (i.e. albite) and a $Ca^{2+}$-rich phase (i.e. bytownite) and assuming that these dissolved independently, the agreement between modelled and observed base cation-ratios were further



improved (model 3). The stronger Si-brake for albite than for bytownite allowed for more $Ca^{2+}$ to be released, and the $Ca^{2+}$/BC-ratio increased steadily throughout the whole simulation period.

The rapid decrease of saturated hydrological conductivity with depth means that the residence time of water also increases

rapidly with depth in the soil. And since the water reaching the stream moves to the stream in the saturated soil below the water table (except at the highest flow rates in these catchments) the age of the water in the stream also increases as flow rates decline (Bishop, 1991; Ameli et al., 2016b). This explains why deeper depths in the soils profile and lower streamflow rates are proxies of water age. The correlation of age to catchment size relates to deeper and longer flow paths in larger catchments (Peralta-Tapia et al., 2015).

### 3.3 Hillslope mass balance calculations

The calculated total flux of base cation release from mineral dissolution of the study hillslope was calculated to be 0.23 eq·m$^{-2}$·yr$^{-1}$, using the original PROFILE equations. With the new equations, the calculated base cation release was about 50 % lower, 0.12 eq·m$^{-2}$·yr$^{-1}$. The latter estimate of base cation release from mineral dissolution is in much better agreement with the

hillslope mass balance (see Erlandsson et al., 2016), especially for $Na^+$ and $K^+$ (Fig. 6). The sinks and sources do not equal each other because of unknown terms in the mass balance, such as net ion exchange, uptake by other plants than trees and precipitation of secondary phases. However, when the original PROFILE equations were applied, the source terms are four times larger than the sink terms for $Na^+$ and nine times higher for $K^+$.

While the median WTT was 2.1 years, the median "age" of monovalent cations, $Na^+$ and $K^+$, was around 3.7 years, i.e., half of the transport of these ions is carried by water older than 3.7 years. For the divalent cations, $Ca^{2+}$ and $Mg^{2+}$, the corresponding age was c:a 5.5 years (Fig. 7). Thus, "old" water is relatively more important for the transport of divalent cations than for monovalent cations.

**4 Discussion**

### 4.1 Functions of the revised PROFILE equations

The PROFILE model was developed in the late 1980's and is still being used in its original form. While the PROFILE equations demand the determination of many parameters, unique for every mineral, a large majority of the parameters found in the original model formulations were based on empirical studies in the form of single mineral lab-studies of dissolution rates. For

these reasons, none of the parameters in the original set of equations were changed in the revision. Instead, a number of features



were added, which are more important in the saturated zone with longer water residence times: The Si-brakes, the OH--term, and the brakes on the $CO_2$-term.

In a catchment perspective, the quantity of weathering products delivered to runoff is a function of the dissolution rate,
integrated over the time the soil water is in contact with the mineral surfaces, and the total mineral surface area. With everything else constant, this produced a nearly linear relationship between base cation concentrations and WTT, using the original PROFILE equations. Adding the Si-brake disrupts the linear proportionality between the rate of base cation export and the increase in residence time or mineral surface area. These revisions, however, do not contradict the applicability of the original PROFILE model to the unsaturated zone. The WTT in the unsaturated zone in shallow till soils may of course vary, but would
typically be around one year or less in Fennoscandia and other areas with similar glacial till soils and climate regimes (Rohde et al., 1996; Amvrosiadi et al., 2017b). The calculated difference in base cation release between the original and the new set of PROFILE equations was relatively small at those time scales; the calculated release from the new equations was 81 % of that from the original equations after one year WTT. It has however been noted in previous PROFILE studies that the release of $Na^+$ is overestimated (Stendahl et al., 2013; Kronnäs et al., 2019), and one cause for this may well be due to neglecting the
retarding effects on dissolution rates from silica.

### 4.2 Base cation rations – similarities and differences between modelled results and observations

In this study we have demonstrated how a model for mineral dissolution kinetics, with a sound theoretical basis in thermodynamics and the transition-state-theory, is able to explain observed patterns of base cation concentration and
distribution in a soil profile and stream water from a poorly buffered glacial till catchment. Ideally, the modelled results would have been evaluated against age-determined groundwater samples along the deeper flow paths; such analyses are however not available at this site. Here, we made use of some consistent patterns in how base cation ratios in field observations ($Ca^{2+}$/BC, $Mg^{2+}$/BC, $Na^+$/BC and $K^+$/BC) varies with proxies of WTT. The comparison between the model and the sampled soil water profile with respect to base cation concentrations and ratios suggests that the length (i.e. depth) scale for the soil profile (0-10
m) and the time scale for the model (0-48 years) are comparable.

The model correctly reproduced most of the observed patterns in base cations-ratios: The initial increase of $Na^+$/BC, followed by a sharp decrease below the groundwater level, and the generally increasing trends of $Ca^{2+}$/BC and $Mg^{2+}$/BC with increasing WTT (Fig. 5). The clear exception is $K^+$/BC, which increased with WTT in field data, especially in the catchment samples,
whereas the modelled $K^+$/BC decreased with WTT due to the strong Si-brake for K-feldspar, which is the main source of $K^+$. There are several possible explanations for the observed increasing $K^+$/BC with catchment size. Two of the more likely are:

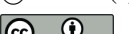



1. $K^+$ is the base cation that is involved most in biological cycling (Likens et al., 1994). It is rapidly taken up and circulated by trees, which is seen in the uppermost part of the profile, where $K^+/BC$ is high in the uppermost layer due to high $K^+$-concentrations in throughfall, but very low in the remainder of the unsaturated zone. As the proportion of $K^+$-poor unsaturated flow to runoff decreases with increasing catchment size, the $K^+/BC$-ratio should be expected to increase.

2. Although the main K-bearing mineral, K-feldspar, should dissolve at a very slow rate in the saturated zone, there is a possibility that $K^+$ is released from the mineral by non-stoichiometric dissolution. This would then be an alteration process rather than a dissolution process. As $K^+$ is replaced with other cat ions, either $Ca^{2+}$ or $Mg^{2+}$, the mineral is gradually altered from K-feldspar to illite or smectite (Sverdrup, 1990). This process would thus form a source of $K^+$ and a sink of $Ca^{2+}$ or $Mg^{2+}$.

As there are indications from field data that $K^+/BC$ increases even in the deep groundwater where there is no biological uptake (the soil profile in the left panel of Fig. 2, and the right part of the in-stream C-Q relationship in the right panel of Fig. 2), we conclude that at least the second hypothesis is likely to be true. Furthermore, as the observed $Mg^{2+}/BC$-ratio does not increase as much as would be expected from the model (the catchment plot in the mid-panel of Fig. 2 and the in-stream C-Q relationship in the right panel of Fig. 2), an exchange between $K^+$ and $Mg^{2+}$ appears to be the dominant mechanism.

### 4.3 Irreversibility of primary mineral dissolution

Most weathering models describe the mineral dissolution rate as a function of chemical affinity, where the dissolution rate approaches zero as the mineral approaches equilibrium with the surrounding soil water solution (Aagaard and Helgeson, 1982). Thus, the dissolution rate described by these models is the net dissolution rate, i.e. the sum of the forward and backward

reaction, which implies that the mineral dissolution reaction is reversible. However, for reactions at low temperature and pressure, the dissolution of primary minerals is irreversible, as the conditions for forming these minerals are simply not present (Sverdrup, 1990; Sverdrup and Warfvinge, 1995).

The PROFILE model adopts a different philosophy, where the dissolution reaction is not assumed to depend on the chemical

affinity. Since no backward reaction is assumed, the PROFILE equations aim to describe only the forward reaction rate, i.e. the pure dissolution reaction. Consequently, these equations allow for minerals to dissolve even if the mineral is apparently saturated with respect to the surrounding soil water solution. As an example, modelled initial dissolution rate for K-feldspar was $4.5 \cdot 10^{-15}$ $mol \cdot m^{-2} \cdot s^{-1}$. After approximately 2.5 months, K-feldspar is apparently in equilibrium with the surrounding solution, i.e. the calculated saturation index = 0. Using the new PROFILE-equations, the K-feldspar is still dissolving at a rate

of $1.5 \cdot 10^{-15}$ $mol \cdot m^{-2} \cdot s^{-1}$, i.e. 33 % of the initial rate. If conditions (i.e. heat and pressure) had allowed for K-feldspar to form, the dissolution reaction would have been counterbalanced by an equally fast precipitation reaction, however, in a shallow soil of low temperature, this reaction is not possible. After one year of WTT, K-feldspar is over-saturated if geochemical



equilibrium is assumed, with a calculated saturation index = 3. According to the new PROFILE equations, the K-feldspar is still dissolving, but at a very low rate of $2.3 \cdot 10^{-16}$ mol·m$^{-2}$·s$^{-1}$, i.e. 5 % of the initial rate.

## 5 Conclusions

5    Revising the mineral dissolution equations of PROFILE is a major step forward in modelling weathering rates on a catchment scale. Our revised equations may now be incorporated into the more complex dynamic ForSAFE model. The PROFILE steady-state approach proposed here is applicable to mineralogically homogenous hillslopes as a tool for estimating element fluxes. Given measured texture, mineralogy and reasonable WTT based on, for example, isotope data, it is a valuable tool for studying hillslope and catchment chemistry in the boreal zone of mature forest systems.

For a more complete description of base cation release from catchments, especially on a longer time perspective, a kinetic description of the gradual transformation from primary to secondary minerals would be useful. While the chemical sequence for such mineral alterations, for example feldspars to illite, or biotite to vermiculite, are known (Sverdrup et al., 2019), the kinetics are not, and this would probably require a calibration approach as well as data from catchments with substantially 15    older water fractions than the ones studied here.

As the revised PROFILE equations otherwise mostly reproduced base cation concentrations and ratios in the saturated zone correctly, the next step is to test these equations in a model that produces stream runoff., thus making the link from the unsaturated zone to groundwater and runoff from an entire catchment. The two-dimensional version of ForSAFE (Zanchi, 20    2016) would allow for this, as this is a dynamic model of biogeochemical cycling that has been developed from PROFILE, and its one-dimensional, steady state conceptualization. This would be a major improvement on the mass-balance approaches currently used to define critical loads for surface waters. These mass balances are highly uncertain when using directly observed inputs and outputs, even with a decade or two of measurements (Simonsson et al., 2015; Akselsson et al., 2019). The mass balance methods also fail to address feedback mechanisms on chemical weathering. For example, tree uptake and 25    deposition affect soil water pH, aluminum and base cation concentrations, all of which effect mineral dissolution rates (Erlandsson et al., 2016). Climate change affects both soil temperature and moisture, which will also impact mineral dissolution (Akselsson et al., 2016). The recent development of a two-dimensional version of ForSAFE in combination with these revised weathering equations would allow for modelling the interactions and feedbacks between trees, soils and runoff, thereby making it possible such processes, including the impact of forestry practices on surface water quality.

30





*Data and code availability.* Data on soil parameters from the Krycklan Catchment study are available from Svartberget open database at https://franklin.vfp.slu.se. The PHREEQC code can be made available upon request to martin.erlandsson.lampa@lansstyrelsen.se.

*Competing interests.* The authors declare that they have no conflict of interest.

*Special issue statement.* This article is part of the special issue "Quantifying weathering rates for sustainable forestry" (BG/SOIL inter-journal SI). It is not associated with a conference.

*Acknowledgments.* This study was funded by the Swedish Research Council Formas (reg. no. 2011-01691) within the strong research environment "Quantifying weathering rates for sustainable forestry (QWARTS)".

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





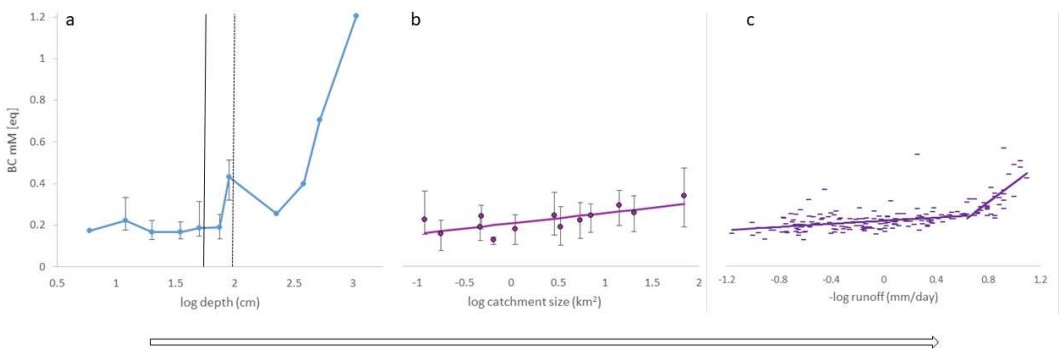

**Figure 1.** Base cation concentrations (mM-eq) plotted against different proxies for water transit time. a) Soil water concentrations in the S22 soil profile (22 m from stream C2), sampled from 6 cm to 10.7 m depth below surface. Uncertainty bars represent the minimum and maximum concentration observed under eight years of sampling (N = 45-69). Vertical lines represent the approximate highest and lowest groundwater surface depth. b) Stream water concentrations in samples from 2016 (N = 19-25) from the Krycklan basin, plotted against subcatchment size (range from 0.12 to 67 km$^2$). Uncertainty bars represent the minimum and maximum concentration observed during sampling. c) Stream water concentrations in samples 2003-2012 from stream C2 (0.12 km$^2$), plotted against the negative logarithm of runoff. Note that stream C2 displayed in c) is the same as the smallest stream in b), but for two different sampling periods.



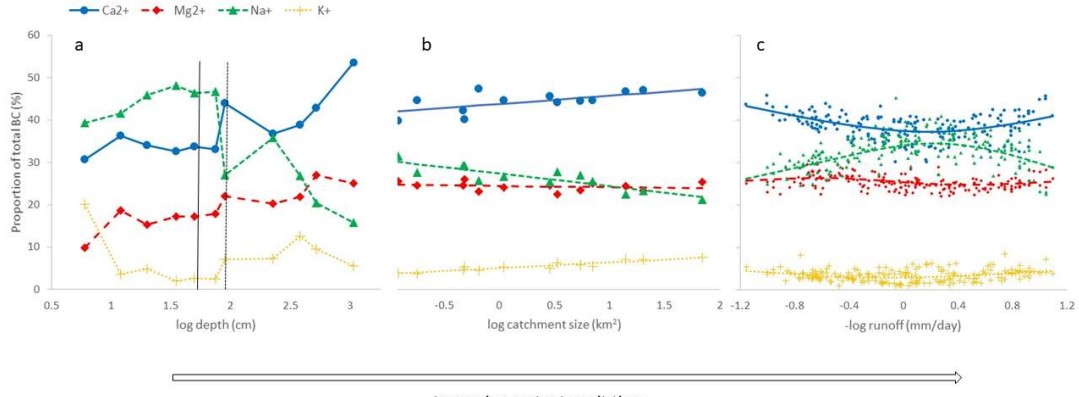

**Figure 2.** Base cation ratios (%) plotted against different proxies for water transit time. a) Ratios in soil water samples from a soil profile 22 m from stream C2, sampled from 6 cm to 10.7 m depth below surface. b) Ratios in stream water samples from 2016 (N = 19-25) from the Krycklan basin, plotted against subcatchment size (range from 0.12 to 67 km$^2$). c) Ratios in stream

5    water samples 2003-2012 from stream C2 (0.12 km$^2$), plotted against the negative logarithm of runoff.



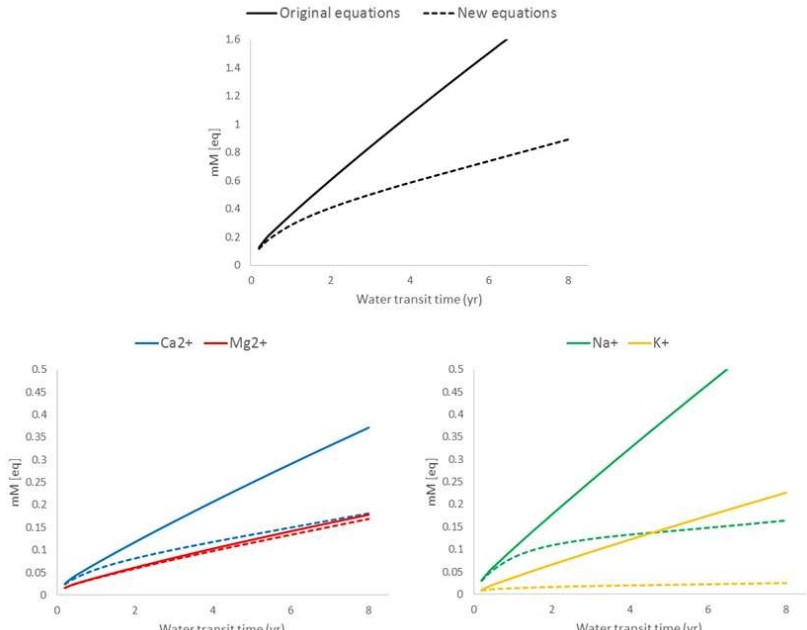

**Figure 3.** Calculated WTT-concentration relationships for the original (solid line) and new (dashed line) PROFILE equations. Upper panel: Total BC-concentrations (black). Lower left panel: $Na^+$ (green) and $K^+$ (yellow). Lower right panel: $Ca^{2+}$ (blue) and $Mg^{2+}$ (red).





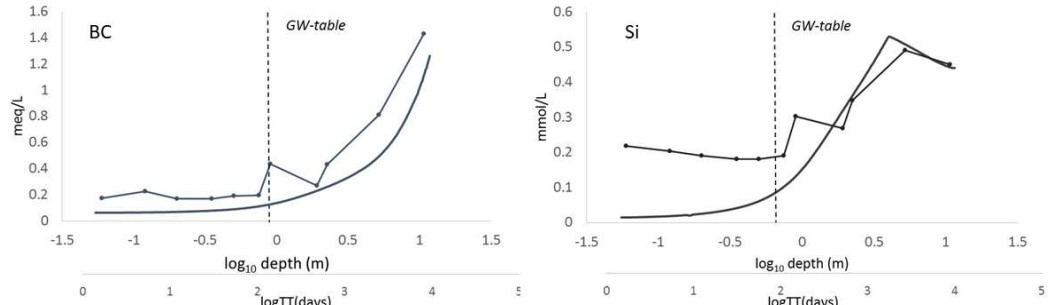

**Figure 4.** Modelled BC (left) and Si (right) concentrations from model 2, plotted against $\log_{10}$ of WTT (smooth lines). Overlain are the observed BC and Si-concentrations from the soil profile, plotted against $\log_{10}$ of soil depth (straight lines with markers). Note that the scales are not directly comparable, instead the figures are intended to illustrate trends and ranges of solute

5  concentrations.



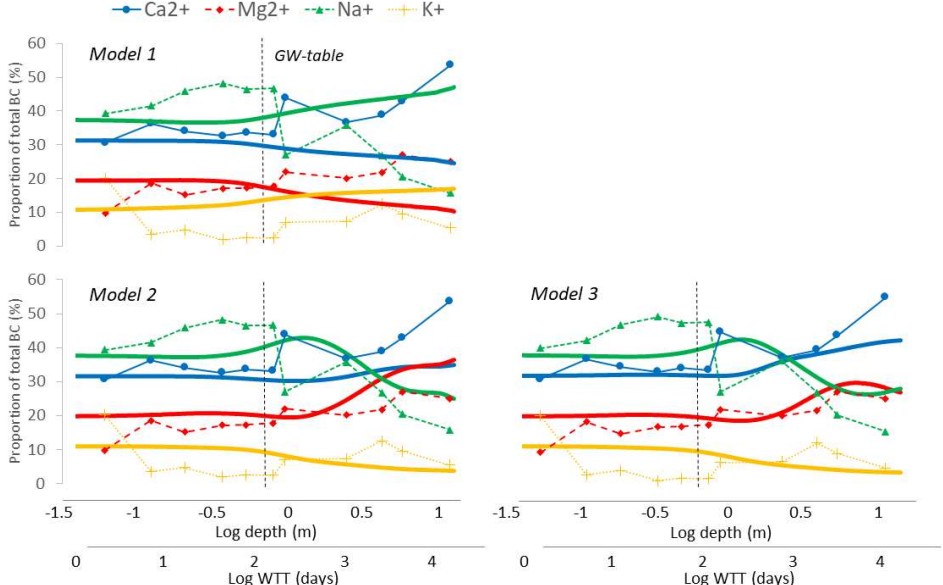

**Figure 5.** Calculated base cation ratios ($Ca^{2+}$/BC, $Mg^{2+}$/BC, $Na^+$/BC and $K^+$/BC) from the three different models plotted against log WTT (thick lines). Model 1 uses the original PROFILE-equations, model 2 the revised PROFILE-equations, and model 3 uses the revised PROFILE-equations with plagioclase split in two phases which dissolves separately. For comparison, the base cation ratios from the soil profile S22 are displayed in the same graphs (markers connected with dashed lines), plotted against the logarithmic depth. Note that the WTT scale is not directly comparable, instead the figures are intended to illustrate trends and ranges of base cation ratios.



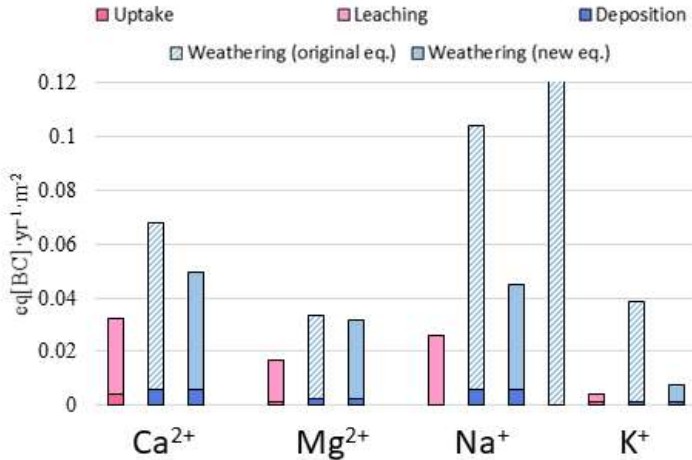

**Figure 6.** Sinks and source terms of the base cation mass balance for a hillslope. The first bar (pink colours) represents sink terms (net tree uptake and runoff), the second bar represent the source terms with weathering calculated from the original PROFILE equations (model 1), and the third bar represent the source terms with weathering calculated from the new PROFILE equations (model 3).




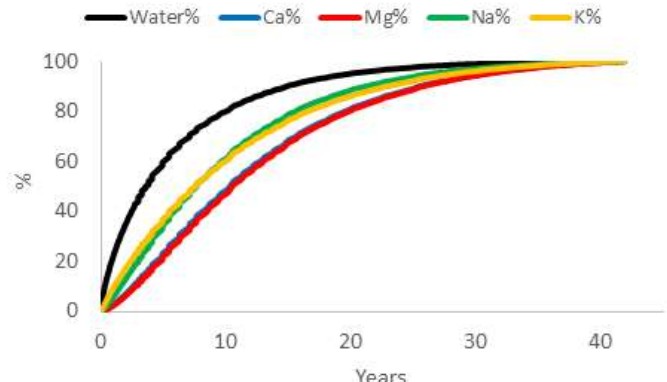

**Figure 7.** Cumulative water transit time distribution (black) of the study hillslope, and the distribution of "age" of the water carrying each of the base cations according to model 3.

