# Peer review of "Catchment export of base cations: Improved mineral dissolution kinetics influence the role of water transit time"

_SOIL, 2019_

## Short Comment (SC1) · 4 Mar 2019

etienne Dambrine

etienne.dambrine@inra.fr

I have left this field since many years, so I may only add two very general comments:

The original model did not take into account the retention of K and Si, as cations and Si were released from primary mineral weathering and non stoechiometric dissolution was not alloowed (mineral transformation, from mica into vermiculite for instance, as well as allophane neoformation). Now, using some kind of dissolution breakes (!), retention is allowed. And this is an improvement !

Profile has been very many times used to model soil solution concentrations and

stream concentrations. This new version predicts, for the study site, that only half of the base cations previously computed with profile are released. I wonder if this proportion can be generalized and how this copes with previously published results;

---

## Referee Comment (RC1) · Anonymous Referee #1 · 15 Mar 2019

It is sometime since I had anything to do with the PROFILE model. I published a series of papers in the 1990s [most notably Hodson et al (1996) Applied Geochemistry 11 835 – 844 and Hodson et al. (1997) Water, Air and Soil Pollution 98 79 – 104] that were critical of the model, not so much in terms of the output, which gives results for weathering rates similar to those determined by other, relevant methods, but the way the model achieves those results. I would argue that any application of the model needs to consider those limitations or at least acknowledge them. I think that this is particular the case in a paper such as this one that: a) involves the original model authors – it is important to acknowledge that researchers independent of the model have raised concerns about it and b) presents an improvement of the model – it is important

to understand the context of that improvement, is the model still using incorrect mineral formulae, reliant on input of relative surface area of different minerals (rather than the typically used relative weight % derived from XRD or normative data), reliant on an unproven formula that calculates total mineral surface area on the basis of soil texture, uses reaction orders and rates that are open to question etc. which are issues raised in the above papers about the model. From the opening of the Discussion it would appear that the original form of the model, complete with disputed parameters was used. This is not to say that the paper should be changed extensively, just that published peer-reviewed journal articles from independent groups that raise issues about this model should at least be acknowledged and / or the concerns raised in those papers addressed.

In the abstract it would be good if in the abstract the authors were able to quantify the degree of improvement in model predictions vs. observations rather than simply stating that there is an improvement.

Around line 26 of the abstract it is stated that the PROFILE equations aren't adapted for the unsaturated zone – I think this is a typo for the saturated zone, consistent with the statement in the introduction around line 3 of the third page where it is stated that the equations are restricted to the unsaturated soil domain.

In section 2 the authors indicate the form of various retardation factors used in the model. Given issues with the derivation of some values used in the model raised in the Hodson et al. papers it would be good if the authors, at least in supplementary information, could indicate how these new retardation factors were derived via plots of the data used in their derivation.

Equation 9 – use of texture to calculate mineral surface area should not be used. There are contradictory statements in the publications of Sverdrup regarding the data which were used to develop this equation, in particular whether soils were treated to remove organic material or organic material and sesquioxides (see Warfvinge, P. and Sverdrup,

H.: 1995, 'Critical Loads of Acidity to Swedish Forest Soils'. Reports in Ecology and Environental Engineering 5, Lund University and Sverdrup, H. U.,Warfvinge, P.: 1995, 'Estimating field weathering rates using laboratory kinetics', in White,A. and Brantley, S. (eds.),Weathering Kinetics of SilicateMinerals,Reviews inMineralogy 31. Min. Soc. of Am.) and the inclusion (or not) of an additional fourth term for coarse sand. In addition, despite citing this equation on numerous occasions the publications of Sverdrup et al. have never published the data used to derive the equation. The only data presented to test this equation is in our paper Hodson et al. 1998 (Hodson, ME, Langan, SJ and Meriau, S (1998) Geoderma 83 35 – 54). My recollection is that in Hodson et al. 1998 there are errors in the units but that issue non-withstanding it is the only published test of the relationship and the relationship was found not to stand. The relationship predicts values which are the same order of magnitude as actual measurements but as such, given the accuracy of mineral weathering calculations, it would be more realistic to use a constant for this term. Given the statement in the discussion that the model has a sound theoretical basis in thermodynamics and Transition state theory I do feel it is important to be open and clear about the derivation of variables like the surface area term (and some of the others highlighted in Hodson et al., 1996, 1997). Alternatively (and better) the authors could finally publish the data used to justify their equation (9) – both the data used to generate the equation and the independent data used to validate it.

After the authors write that they use this equation they then go on to write that it gave a number which they felt was too high so they used a lower value similar to values used elsewhere in the PROFILE. To me this seems like having ones cake and eating it. Either the equation is applicable and should be used through out or it isn't applicable and shouldn't be used. By setting the value at an arbitrary level surely surface area is being used as a fitting parameter not an input calculated for the horizon in question.

More generally it would be helpful if a list of the input parameters used in the model were provided in Supplementary information.

As stated in the Discussion (4.3) the proposed modifications are welcome as they help to address the issue regarding chemical affinity where previously mineral phases could continue to dissolve even when the predicted solution concentration was saturated with respect to that mineral. I would suggest that, for the example given for example where the dissolution rate of K-feldspar decreases by an order of magnitude or so after the solution is saturated, this is an improvement but it would be better if the model were modified so as to predict no dissolution (or net dissolution) in a saturated solution which is surely more realistic. That being the case it would be useful if the authors could explain why they chose to modify their model in a way that acknowledges and addresses this issue to a certain extent but not fully.

In the conclusions the authors state that this version of the model is applicable to mineralogically homogeneous hillslopes. This limitation is presumably because of the complexity of considering movement of packages of cation laden water from one mineralogical environment to another where retardation of reactions will change. However it might be useful for the authors to offer an opinion on how "homogeneous" soils have to be for the model to be OK for use. Soils are very inhomogenous on a number of different scales.

In summary, this paper represents a useful advance of the PROFILE model. However, it fails to acknowledge independent, published concerns regarding the parameters used to drive the model. The uncertainty inherent in calculations of this nature do result in the model generally performing quite well when compared to solution chemistry data sets.

In general there are minor typos / issues of grammar that need addressing.

---

## Referee Comment (RC2) · Anonymous Referee #2 · 20 Aug 2019

Introduction: The text is rather heavy reading because of the complexity of the sentences, non-the less it is intelligent and proves deep insight, but needs some extra effort on the pedagogical side. The English language is flawless and reads well, but gets a bit complex, which I am sure can be abated.

For helping the non-expert reader I suggest you make a couple of simple conceptual drawings of your systems at the different scales (catchment, the hill-slope from Krycklan or micro site etc.) where you visualize with some examples, arrows or small 'ratios' the terms: 'runoff', 'water transit time', 'proxies for WTT', 'discharge', 'base cation release, -flow and flux', 'dissolution of minerals', 'weathering rates', 'chemical

weathering', 'biological uptake', 'solutes', 'pore water', 'soil water', 'mineral surfaces', 'steady state', 'non-interacting', 'pathway', 'water parcel', 'stream concentration', 'mineral dissolution rate', 'concentration – discharge relasionship', 'chemostatic behaviour', 'secondary mineral formation', 'rooting zone', 'unsaturated soil', 'saturated zone', 'vertical flow', 'water table', 'the silica effect', 'ratios of different important base cations Ca, Mg, Na, K'; a more clear overview of these terms would ease the 'putting into context' of this magnificent modelling work.

Once this is done the text needs probably a bit of refinement to simplify the many terms and split some complex sentences but this will become clear while working on that diagram.

Please clarify: The 'Glacial till soils' partly gets get separate introduction and soil is also generalized as unsaturated and saturated soils, could you rephrase or simply explain why this type has this focus and if it is included as saturated and unsaturated ? The minerals: albite, bytownite, could you explain what these are with a little detail?

Site description: is the coniferous forest a managed forest ? if so, what age is the forest ? has it been disturbed with harvesting within recent years ? Is the soil saturated / unsaturated or variable ?

Discussion: In order to get a better overview of the improvements with PROFILE, could you make a Table with the improved features, if these are new additions and the outcome (effect). You might need to fish out information from the connecting paper Sverdrup et al 2019 for the OH brake.

You need to update the reference to Sverdrup et al 'this issue' 2019; both in the text and in the reference list. Also you will need to provide the mentioned equations as supplementary material to this paper in SOIL, if it is not already published or has another current publication history. This will take some revision time, but should be allowed.

Please also note the supplement to this comment:
https://www.soil-discuss.net/soil-2019-3/soil-2019-3-RC2-supplement.pdf

---

## Author Comment (AC2) · 17 Sep 2019

Introduction: The text is rather heavy reading because of the complexity of the sentences, non-the less it is intelligent and proves deep insight, but needs some extra effort on the pedagogical side. The English language is flawless and reads well, but gets a bit complex, which I am sure can be abated. For helping the non-expert reader I suggest you make a couple of simple conceptual drawings of your systems at the different scales (catchment, the hill-slope from Krycklan or micro site etc.) where you visualize with some examples, arrows or small 'ratios' the terms: 'runoff', 'water transit time', 'proxies for WTT', 'discharge', 'base cation release, -flow and flux', 'dissolution of

minerals', 'weathering rates', 'chemical weathering', 'biological uptake', 'solutes', 'pore water', 'soil water', 'mineral surfaces', 'steady state', 'non-interacting', 'pathway', 'water parcel', 'stream concentration', 'mineral dissolution rate', 'concentration–discharge relasionship', 'chemostatic behaviour', 'secondary mineral formation', 'rooting zone', 'unsaturated soil', 'saturated zone', 'vertical flow', 'water table', 'the silica effect', 'ratios of different important base cations Ca, Mg, Na, K'; a more clear overview of these terms would ease the 'putting into context' of this magnificent modelling work. Once this is done the text needs probably a bit of refinement to simplify the many terms and split some complex sentences but this will become clear while working on that diagram.

* First, we thank the reviewer for the kind words. Illustrating all these terms in figures is quite demanding for a research paper, but we will include conceptual figures to explain at least the core concepts of the paper.

Please clarify: The 'Glacial till soils' partly gets get separate introduction and soil is also generalized as unsaturated and saturated soils, could you rephrase or simply explain why this type has this focus and if it is included as saturated and unsaturated ?

*We will include a short description of glacial tills.

The minerals: albite, bytownite, could you explain what these are with a little detail?

*This will be done in the revision.

Site description: is the coniferous forest a managed forest ?  if so, what age is the forest ? has it been disturbed with harvesting within recent years ? Is the soil saturated / unsaturated or variable?

* Answers will be provided in the revision.

Discussion: In order to get a better overview of the improvements with PROFILE, could you make a Table with the improved features, if these are new additions and the outcome (effect).  You might need to fish out information from the connecting paper

Sverdrup et al 2019 for the OH brake.

* This is a good idea which we will adopt for the revised paper.

You need to update the reference to Sverdrup et al 'this issue' 2019; both in the text and in the reference list. Also you will need to provide the mentioned equations as supplementary material to this paper in SOIL, if it is not already published or has another current publication history. This will take some revision time, but should be allowed.

* Equations are provided in the companion paper in the same issue: Sverdrup et al., "Reviews and syntheses: Weathering of silicate minerals in soils and watersheds: Parameterization of the weathering kinetics module in the PROFILE and ForSAFE models".

---

## Author Response (AR1)

**Point-by-point response**

(page and line references refer to the file without tracking)

**Review #1**

It is sometime since I had anything to do with the PROFILE model. I published a series of papers in the 1990s
5 [most notably Hodson et al (1996) Applied Geochemistry 11 835 – 844 and Hodson et al. (1997) Water, Air and Soil Pollution 98 79 – 104] that were critical of the model, not so much in terms of the output, which gives results for weathering rates similar to those determined by other, relevant methods, but the way the model achieves those results. I would argue that any application of the model needs to consider those limitations or at least acknowledge them. I think that this is particular the case in a paper such as this one that: a) involves the
10 original model authors – it is important to acknowledge that researchers independent of the model have raised concerns about it and b) presents an improvement of the model – it is important to understand the context of that improvement, is the model still using incorrect mineral formulae, reliant on input of relative surface area of different minerals (rather than the typically used relative weight % derived from XRD or normative data), reliant on an unproven formula that calculates total mineral surface area on the basis of soil texture, uses reaction
15 orders and rates that are open to question etc. which are issues raised in the above papers about the model. From the opening of the Discussion it would appear that the original form of the model, complete with disputed parameters was used. This is not to say that the paper should be changed extensively, just that published peer-reviewed journal articles from independent groups that raise issues about this model should at least be acknowledged and / or the concerns raised in those papers addressed.

20 • *We agree with the reviewer on this and have included a discussion addressing these concerns (P 12, L 3-10).*

In the abstract it would be good if in the abstract the authors were able to quantify the degree of improvement in model predictions vs. observations rather than simply stating that there is an improvement. Around line 26 of
25 the abstract it is stated that the PROFILE equations aren't adapted for the unsaturated zone – I think this is a typo for the saturated zone, consistent with the statement in the introduction around line 3 of the third page where it is stated that the equations are restricted to the unsaturated soil domain.

• *The typo has been corrected. However, we would argue that the results were summarized in sufficient detail in the abstract already.*

In section 2 the authors indicate the form of various retardation factors used in the model. Given issues with the derivation of some values used in the model raised in the Hodson et al. papers it would be good if the authors, at least in supplementary information, could indicate how these new retardation factors were derived via plots of the data used in their derivation.

- *This is addressed in the companion paper in the same issue: Sverdrup et al., "Reviews and syntheses: Weathering of silicate minerals in soils and watersheds: Parameterization of the weathering kinetics module in the PROFILE and ForSAFE models".*

5   Equation 9–use of texture to calculate mineral surface area should not be used. There are contradictory statements in the publications of Sverdrup regarding the data which were used to develop this equation, in particular whether soils were treated to remove organic material or organic material and sesquioxides (see Warfvinge,P .andSverdrup,H.: 1995, 'Critical Loads of Acidity to Swedish Forest Soils'. Reports in Ecology and Environental Engineering 5, Lund University and Sverdrup, H. 'Estimating field weathering rates using laboratory
10   kinetics' ,in White, A. and Brantley, S. (eds.),Weathering Kinetics of Silicate Minerals, Reviews in Mineralogy 31. Min. Soc. of Am.) and the inclusion (or not) of an additional fourth term for coarse sand. In addition, despite citing this equation on numerous occasions the publications of Sverdrup et al. have never published the data used to derive the equation. The only data presented to test this equation is in our paper Hodson et al. 1998 (Hodson, ME, Langan, SJ and Meriau, S (1998) Geoderma 83 35 – 54). My recollection is that in Hodson et al.
15   1998 there are errors in the units but that issue non-withstanding it is the only published test of the relationship and the relationship was found not to stand. The relationship predicts values which are the same order of magnitude as actual measurements but as such, given the accuracy of mineral weathering calculations, it would be more realistic to use a constant for this term. Given the statement in the discussion that the model has a sound theoretical basis in thermodynamics and Transition state theory I do feel it is important to be open and
20   clear about the derivation of variables like the surface area term (and some of the others highlighted in Hodson et al., 1996, 1997). Alternatively (and better) the authors could finally publish the data used to justify their equation (9) – both the data used to generate the equation and the independent data used to validate it.

- *While the reviewer's criticism of how mineral surface area is calculated in PROFILE may be justified, we feel that it falls outside the scope of this paper. Our focus is to demonstrate how the new rate laws for*
25   *mineral dissolution described in the companion paper by Sverdrup et al., produce results which are consistent with field observations, especially in comparison with previously used rate laws. Mineral surface area has very little bearing on BC-ratios presented in chapter 3.2.3/Fig. 7. It does influence the calculations of the total BC-release from the hillslope (chapter 3.3/Fig. 8), but still the results are not very sensitive to mineral surface area, and even with a doubling of the parameter value, we still achieve*
30   *realistic estimates. We have included a discussion addressing this (P 12, L 10-14).*

After the authors write that they use this equation they then go on to write that it gave a number which they felt was too high so they used a lower value similar to values used elsewhere in the PROFILE. To me this seems like having ones cake and eating it. Either the equation is applicable and should be used through out or it isn't
35   applicable and shouldn't be used. By setting the value at an arbitrary level surely surface area is being used as a fitting parameter not an input calculated for the horizon in question. More generally it would be helpful if a list of the input parameters used in the model were provided in Supplementary information.

- *It is understandable that the reviewer is skeptical here. However, no model fitting was done, we were simply trying to come up with realistic estimates of soil properties that could represent the upslope mineral soil. Had the parameter been fitted, we would have ended up using a larger number for mineral surface area to get a better agreement with observed data (Figs 6 and 8). In the revised manuscript, we have used the more sound assumption of using average values from the whole soil profile (P 7, L 2-11)*

As stated in the Discussion (4.3) the proposed modifications are welcome as they help to address the issue regarding chemical affinity where previously mineral phases could continue to dissolve even when the predicted solution concentration was saturated with respect to that mineral. I would suggest that, for the example given for example where the dissolution rate of K-feldspar decreases by an order of magnitude or so after the solution is saturated, this is an improvement but it would be better if the model were modified so as to predict no dissolution (or net dissolution) in a saturated solution which is surely more realistic. That being the case it would be useful if the authors could explain why they chose to modify their model in a way that acknowledges and addresses this issue to a certain extent but not fully.

- *This is already explained in a subchapter in the discussion (4.3)*

In the conclusions the authors state that this version of the model is applicable to mineralogically homogeneous hillslopes. This limitation is presumably because of the complexity of considering movement of packages of cation laden water from one mineralogical environment to another where retardation of reactions will change. However it might be useful for the authors to offer an opinion on how "homogeneous" soils have to be for the model to be OK for use. Soils are very inhomogenous on a number of different scales.

- *This has been clarified in the revised paper (P 14, L 25-26).*

In summary, this paper represents a useful advance of the PROFILE model. However, it fails to acknowledge independent, published concerns regarding the parameters used to drive the model. The uncertainty inherent in calculations of this nature do result in the model generally performing quite well when compared to solution chemistry data sets. In general there are minor typos / issues of grammar that need addressing.

**Review #2**

Introduction: The text is rather heavy reading because of the complexity of the sentences, non-the less it is intelligent and proves deep insight, but needs some extra effort on the pedagogical side. The English language is flawless and reads well, but gets a bit complex, which I am sure can be abated. For helping the non-expert reader I suggest you make a couple of simple conceptual drawings of your systems at the different scales (catchment, the hill-slope from Krycklan or micro site etc.) where you visualize with some examples, arrows or small 'ratios' the terms: 'runoff', 'water transit time', 'proxies for WTT', 'discharge', 'base cation release, -flow

and flux', 'dissolution of minerals', 'weathering rates', 'chemical weathering', 'biological uptake', 'solutes', 'pore water', 'soil water', 'mineral surfaces', 'steady state', 'non-interacting', 'pathway', 'water parcel', 'stream concentration', 'mineral dissolution rate', 'concentration–discharge relasionship', 'chemostatic behaviour', 'secondary mineral formation', 'rooting zone', 'unsaturated soil', 'saturated zone', 'vertical flow', 'water table', 'the silica effect', 'ratios of different important base cations Ca, Mg, Na, K'; a more clear overview of these terms would ease the 'putting into context' of this magnificent modelling work. Once this is done the text needs probably a bit of refinement to simplify the many terms and split some complex sentences but this will become clear while working on that diagram.

- *First, we thank the reviewer for the kind words. We have included two additional figures, one illustrating interactions and feedback loops between factors influencing mineral dissolution (Figure 1), and one schematic drawing of the hillslope (Figure 2).*

Please clarify: The 'Glacial till soils' partly gets get separate introduction and soil is also generalized as unsaturated and saturated soils, could you rephrase or simply explain why this type has this focus and if it is included as saturated and unsaturated ?

- *We have included a short description of glacial tills (P 3, L 13).*

The minerals: albite, bytownite, could you explain what these are with a little detail?

- *This has been included in the revised manuscript (P 8, L 4-6).*

Site description: is the coniferous forest a managed forest ? if so, what age is the forest ? has it been disturbed with harvesting within recent years ? Is the soil saturated / unsaturated or variable?

- *Answers have been provided in the revised manuscript (P 6, L 28-30).*

Discussion: In order to get a better overview of the improvements with PROFILE, could you make a Table with the improved features, if these are new additions and the outcome (effect). You might need to fish out information from the connecting paper Sverdrup et al 2019 for the OH brake.

- *We have included this information as text (P 12, L 20-22).*

You need to update the reference to Sverdrup et al 'this issue' 2019; both in the text and in the reference list. Also you will need to provide the mentioned equations as supplementary material to this paper in SOIL, if it is not already published or has another current publication history. This will take some revision time, but should be allowed.

- *Equations are provided in the companion paper in the same issue: Sverdrup et al., "Reviews and syntheses: Weathering of silicate minerals in soils and watersheds: Parameterization of the weathering kinetics module in the PROFILE and ForSAFE models".*

**List of significant changes**

- The model has been rerun using new input data. All soil properties are now taken as average values from the whole soil profile. Numbers and figures have been updated accordingly.
- Two new figures (Figs. 1 and 2) have been included to help the reader to understand some of the key terms used in the manuscript
- A discussion have been included, addressing questions raised regarding calculations of mineral surface area and how results are affected.
- Information on the relative importance of each of the new additions to the PROFILE equations has been added.
- All minor points raised by the reviewers have been addressed.

[revised manuscript text omitted]